# Estimation of Groundwater Recharge in the Lobo Catchment (Central-Western Region of Côte d'Ivoire)

**Kouadio Kouamé Jean Olivier** [1,2,*], **Dibi Brou** [1], **Mangoua oi Mangoua Jules** [1], **Eblin Sampah Georges** [1], **Paran Frédéric** [2] and **Graillot Didier** [2]

1   Laboratory of Environmental Sciences and Technologies, Jean Lorougnon Guede University, Daloa BP 150, Côte d'Ivoire; dibrou2003@yahoo.fr (D.B.); mjul_2@yahoo.fr (M.o.M.J.); sageblin@gmail.com (E.S.G.)
2   UMR 5600, École Supérieure des Mines de Saint-Étienne, 158 Cours Fauriel, 42023 Saint-Étienne, France; paran@emse.fr (P.F.); graillot@emse.fr (G.D.)
*   Correspondence: olivierkouame05@gmail.com; Tel.: +225-0701162425 or +33-6-05797156

**Abstract:** Determination of groundwater recharge is a major challenge in areas where rainfall is generally abundant. Variability and uncertainty are inherent in the estimation of recharge, and several methods are therefore recommended for its estimation at a regional level. In this study, we evaluated several methods for estimating recharge: the web GIS-based automated hydrological analysis tool (WHAT), water table fluctuation (WTF), hydrograph analyses, a recession curve displacement method, graphical separation, and empirical formulas. The annual recharge estimated by combining direct recharge and base-flow varied from 84 mm in 2019 to 66.4 mm in 2020. The mean direct recharge was about 44 mm in 2018 and 57.3 mm in 2019, representing about 4% and 5% of the respective rainfall. In 2020, this direct recharge was 43 mm, or about 6% of rainfall, around 25% lower than in 2019. Base-flow separation methods and recession curve displacement generally gave low results, whereas modified empirical formulas gave results close to those of the WTF method and were considered more consistent and reasonable. The regression curve displacement method implemented in the United States geologic survey (USGS) RORA program was found to be unsuitable for the study area. However, the other methods presented more reasonable results and could be used to estimate groundwater recharge in the study area.

**Keywords:** hydrograph analysis; recession curve displacement; piezometric fluctuation; empirical formula; base flow separation method

## 1. Introduction

Climate change and population growth are putting considerable pressure on the water resources of the Lobo catchment area, which are mainly used to supply drinking water to the population of the city of Daloa. Located in the central-western region, Daloa is the third-largest city in Cote d'Ivoire and forms an economic hub for all localities in the Lobo catchment. The city and its surroundings are supplied with drinking water from the Lobo reservoir. This resource is crucial for these localities but is currently threatened by overexploitation and siltation [1] and is highly eutrophic due to multiple sources of pollution. Due to the poor quality of the water supplied to the population, repeated interruptions in supply, overexploitation, and reduction in inflow to the river [1], exploitation of groundwater is necessary to improve the supply of drinking water. However, predicting the duration of possible exploitation and the flow associated with it requires an assessment of the renewal of the resource, i.e., the recharge.

Groundwater recharge is one of the most important processes for the sustainable management of groundwater [2]. Under natural conditions, recharge is equal to discharge [3], and groundwater is therefore balanced. To ensure sustainable management of groundwater resources, their exploitation should not exceed recharge levels [4]. However, any

usage will come at the cost of lower discharge levels, which will have a downstream effect on indirect recharge. The estimation of direct groundwater recharge is very difficult [5], and it is generally recommended that several methods should be applied and the results compared between them to avoid the error and uncertainties inherent to each method [6,7]. Several methods can be used to determine groundwater recharge, including direct measurements in the unsaturated zone based on aquifer characteristics (porosity and permeability), fluctuations in water level due to direct recharge [8,9], and groundwater flow (and transport) models [10]. River hydrograph analyses such as base-flow separation, recession curve displacement, and those using GIS-based, web-based computer code (an automated hydrological analysis tool) are based on the fact that groundwater feeds rivers under wet climate conditions [10,11]. In addition, groundwater recharge can be estimated by comparing the concentration of chloride in rainwater and groundwater [12]. For areas with insufficient data, such as the Lobo catchment, groundwater recharge can be calculated using empirical formulas based only on the rainfall amount [10,13–15]. The suitability of each method depends on the climate, the characteristics of the catchment, data availability, and objectives.

The purpose of this study is to estimate the recharge of the water tables of the Lobo catchment area to improve the management of this resource. In this study, recharge was determined with the WTF method and compared with hydrograph analyses, recession curve displacement, base flow separation, and empirical formulas.

## 2. Study Area

The Lobo catchment area is located in the central-western region of Cote d'Ivoire, between 6°05′ and 6°55′ west longitude and between 6°02′ and 7°55′ north latitude (Figure 1). The area is limited to the regions of upper Sassandra, whose capital city is Daloa and is part of the Worodougou region (Seguela). Daloa forms the economic hub of the area. This catchment has an area of 7000 km². The mean rainfall over the period 1971–2016 was about 1200 mm, and the average temperature was 25 °C. This figure also shows all the observation points and gauging stations in the Lobo catchment area. The observation points have made it possible to collect piezometric data over three years (2018–2020), while the gauging stations have made it possible to obtain hydrological data over two years (2019–2020).

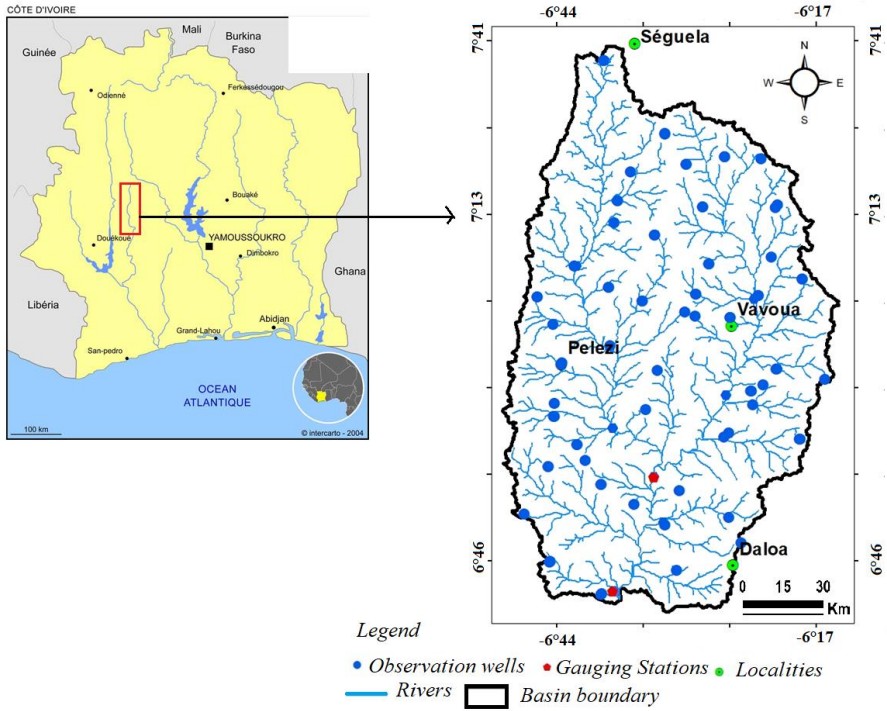

**Figure 1.** The Lobo catchment in Nibéhibé.

The Lobo catchment area as a whole consists of plains and plateaus. The plains vary in altitude from 160 to 245 m and are located in the southern part of the catchment area. The plateaus are located at altitudes ranging from 245 to 480 m and occupy most of the Lobo catchment. The population is estimated at 1,103,059 inhabitants [16], giving a density of 165.67 inhabitants per km$^2$, with an annual growth rate of 3.72%. Within this zone, the drinking water supply for large agglomerations is ensured by the collection of surface water, whereas in rural areas, drinking water comes mainly from boreholes.

### 2.1. Stream Runoff

Figure 2 illustrates the variation in average monthly flows at the gauging stations of Sikaboutou and Nibéhibé between 2019 and 2020. Flows are high between July and October, which corresponds to the main rainy season in the area, and are very low between December and February, which corresponds to the dry season in the area. The mean flows at these stations varied from 9 to 10 m$^3$/s, with a maximum of 39 m$^3$/s recorded in September 2019, and from 5 to 7 m$^3$/s with a maximum of 25 m$^3$/s recorded in October 2020. These flows are generally higher at the Sikaboutou station upstream of the Lobo reservoir than at the Nibéhibé station at the outlet of the Lobo catchment and downstream of the Lobo reservoir.

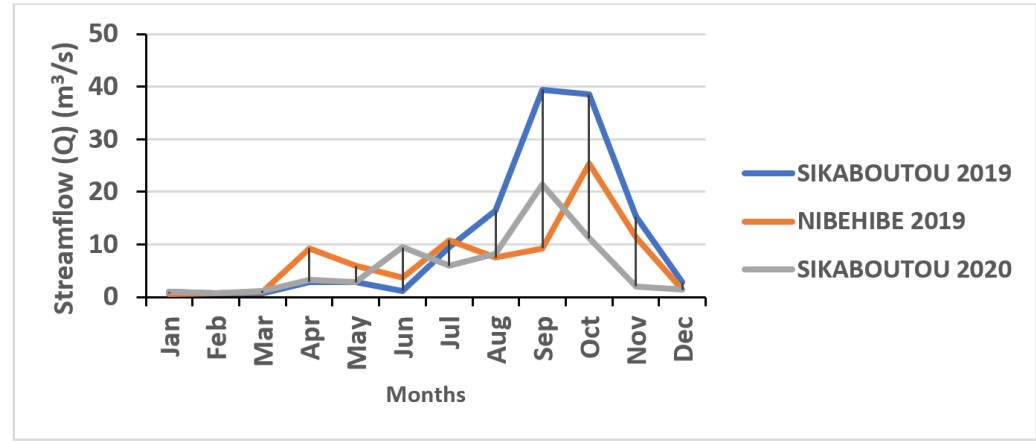

**Figure 2.** Variation in the average monthly flows at Sikaboutou and Nibéhibé gauging stations between 2019 and 2020.

### 2.2. Geological and Hydrogeological Overview of the Study Area

The geological formations of the basin belong mainly to the Precambrian basement (Middle Precambrian) and are grouped into two main entities: magmatic rocks that are mainly composed of granite and metamorphic rocks composed of schist [17]. The magmatic rocks encountered are plutonic and volcanic types. They are essentially made up of the granitoids that are found throughout most of the catchment area. In the study area, migmatites and schists are geological formations that represent metamorphic rocks lodged mainly in the riverbeds (Figure 3). In the Lobo catchment area, a composite aquifer is formed by weathering aquifers (superficial) and fractured aquifers (deeper). Saprolite aquifers develop in sandy clay formations and granitic arenas. Fissured aquifers underlying weathering aquifers form important reservoirs [18].

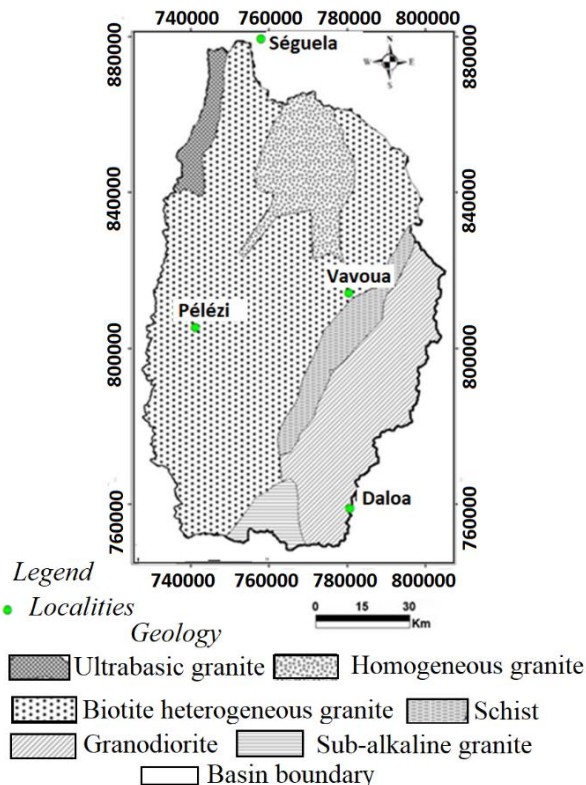

**Figure 3.** Geological formations of the Lobo catchment modified from [19].

*2.3. Groundwater Recharge Processes*

Groundwater recharge depends on several factors, including the geologic area, its topography, and the hydroclimatic conditions of the area. Generally, the largest part of the rainfall over a catchment returns to the atmosphere through evapotranspiration [20]. Within a catchment area, three forms of recharge contribute to the recharge process: localized recharge (joints, depressions, rivulets), indirect recharge (rivers), and direct recharge (precipitation) (Figure 4).

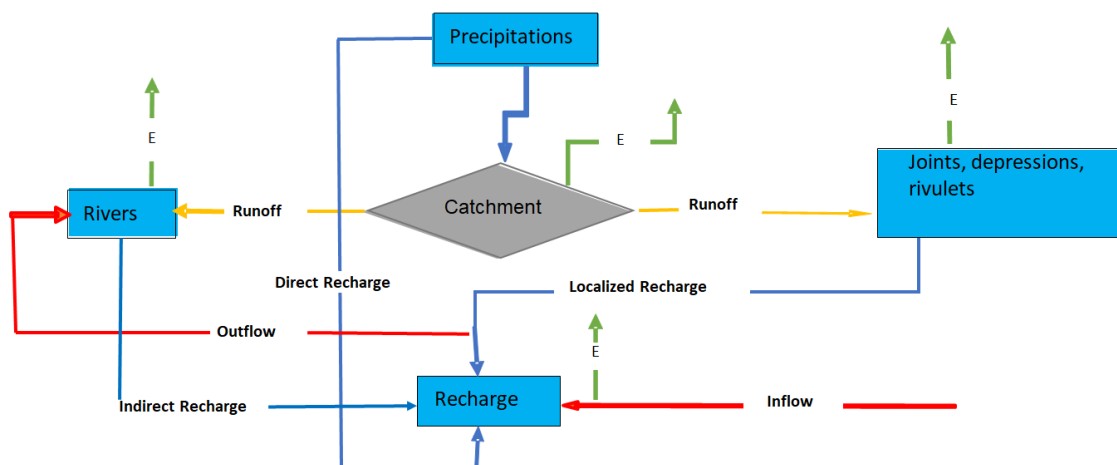

**Figure 4.** Description of the groundwater recharge mechanism in a catchment, modified from [6].

## 3. Materials and Methods

*3.1. Data*

The data used in this study consist of piezometric data with seasonal time steps over the period 2019–2020 and rainfall data with daily time steps over the period 2000–2020.

These data were collected at Daloa synoptic station. Hydrometric data cover the periods 2019 for the Nibéhibé station and 2019–2020 for the Sikaboutou station.

*3.2. Methods*

In this study, recharge was first estimated by methods based on the analysis of river hydrographs. Then, direct recharge was also estimated by the WTF method and empirical methods, and later, by combining the direct recharge estimated by the WTF method and the base-flow estimated by the WHAT method, the annual recharge was estimated (Figure 5).

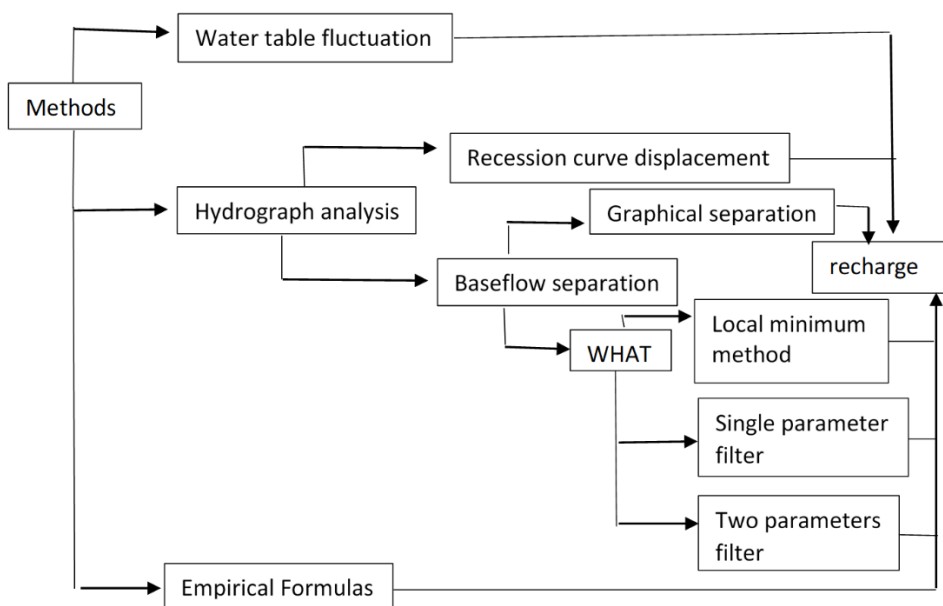

**Figure 5.** The set of methods used to calculate groundwater recharge in the Lobo catchment.

3.2.1. Hydrograph Analysis

Recession Curve Displacement

The RECESS program [21] was used to determine the master recession curve (MRC) of stream-flow records for the period 2019–2020, when all flow can be considered groundwater discharge. The program uses a cyclic interactive procedure for selecting several periods of continuous recession (recession segments; Figure 6) and determines a best-fit equation for the rate of recession as a function of the logarithm of flow. Then, the coefficients of this equation are used to derive the MRC, and finally, to determine the recession index (K), which represents the recession rate in days per log cycle. The critical time Tc, representing the time at which the recession becomes linear, can be determined from Equation (1). The detailed procedure for this process is given in [21].

$$R = \frac{2(Q_2 - Q_1)K_I}{2.3026} \tag{1}$$

where R is the recharge for the streamflow peak ($m^3$), $Q_1$ is the groundwater discharge at the critical time (Tc) ($m^3$/day), and $Q_2$ is the groundwater discharge at Tc ($m^3$/day).

The critical time $T_c$, representing the time at which the recession becomes linear, can be determined from Equation (2). A detailed procedure for this process is given by [22]:

$$T_c = 0.214 \, K_I \tag{2}$$

where $K_I$ is the recession rate in days per logarithmic cycle and can be estimated manually [9,23]. In this study, the United States Geologic Survey RORA software was used to apply this method.

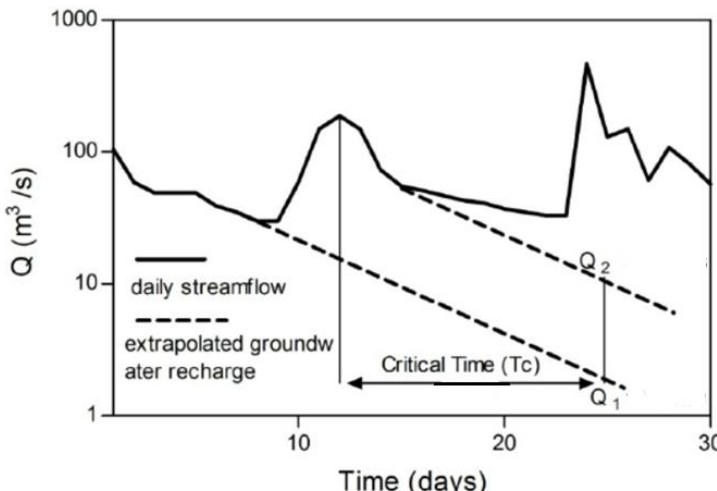

**Figure 6.** Procedure for the displacement recession curve method (Q is the streamflow).

Graphical Separation (Constant Slope Method)

Based on the assumption that direct and hypodermic flows (simply called runoff) are combined, linear methods allow the separation of runoff from base-flow by identifying the start and end points of runoff directly on the hydrograph. This method involves joining points A and B by a straight line, as in Figure 7. Point A corresponds to the beginning of a rising phase of the hydrograph, and point B corresponds to the inflection point of the receding phase. This point is determined by the intersection formed after extending the recession and drying curves, which are linear when a logarithmic transformation of the ordinate axis is applied. The base-flow rate is calculated by connecting the starting point of a rainfall event to the point of inflection on the declining part of the event, with the area under the curve representing the base-flow rate [24].

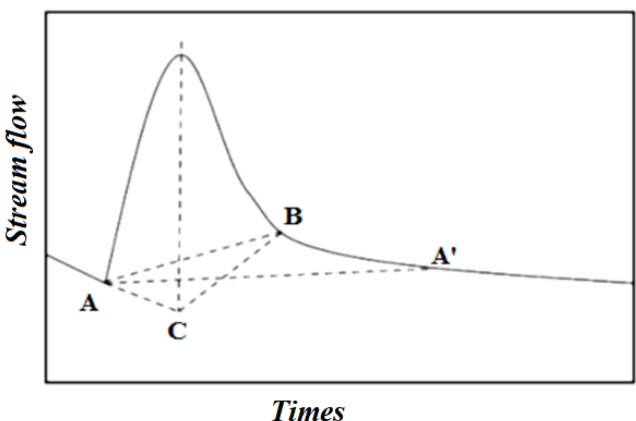

**Figure 7.** Linear separation method for hydrograph components [25].

Automated Web GIS-Based Hydrograph Analysis Tool (WHAT)

Ref. [11] developed the WHAT tool that includes three base-flow separation techniques: the local minimum method, a single-parameter digital filter, and two-parameter digital filters. The digital filters are used to divide the stream-flow hydrograph into high-frequency (direct runoff) and low-frequency (base-flow) components.

✔ Single parameter digital filter [26]

In the first step, the direct stream runoff is assumed to be zero ($R_0 = 0$), and the base-flow is equal to the streamflow ($B_0 = Q_0$,). Thus, for each step, the runoff is calculated from Equation (3):

$$R_p = \propto R_{p-1} + \frac{(1+ \propto)}{2}(Q_P - Q_{P-1}) \tag{3}$$

where $Q_p$ is the streamflow at time step p, $R_p$ is the direct runoff at time step p, and $\alpha$ is the base-flow filter parameter. The following assumptions are made: if $R_p < 0$, then $R_p = 0$, or if $R_p > Q_p$, then $R_p = Q_p$. Finally, the base-flow is calculated from Equation (4):

$$B_P = Q_P - R_P \tag{4}$$

where $B_P$ is the base-flow at time step p.

✔ Two-parameter digital filters or Eckhardt method [27]

Using the same procedure as for the single-parameter numerical filter, the direct runoff is also considered zero ($R_0 = 0$), and the base-flow is equal to the streamflow (**$B_0 = Q_0$**). Then, at each step, the runoff is calculated using Equation (5):

$$B_p = \frac{(1 - \mathrm{BFI}_{max}). \propto .B_p + (1- \propto).\mathrm{BFI}_{max}.Q_p}{1- \propto .\mathrm{BFI}_{max}} \tag{5}$$

with $\alpha$ representing the filtering parameter of the base-flow. $\mathrm{BFI}_{max}$ is the maximum value of the long-term ratio of base-flow to total streamflow. This ratio is equal to 0.25 for perennial streams with hard rock aquifers, 0.50 for ephemeral streams with porous aquifers, and 0.80 for perennial streams with porous aquifers. For our study, the ratio of 0.25 for perennial streams was used.

✔ Local minimum method

In this approach, each daily flow is checked to see whether or not it is the lowest flow over half the interval minus one day (0.5 [2N−1]) days) before and after the day tested [28]. If this is the case, it is considered a local minimum flow. Then, by crossing all local minimum points with a straight line, we obtain the base-flow [28]. Ref. [29] recommended that this method be applied in the dry season due to overestimation of its results during the rainy season. Within a year, these base-flows can also correspond to the two lowest monthly base-flows [30]. Following the recommendations of [31], the stable base-flow can be calculated as follows.

- Use the methods of graphical separation, local minimum, and one- and two-parameter numerical filters to calculate the monthly base-flow.
- Calculate the average monthly base-flow over a long period.
- Sort and accumulate the long-term average monthly base-flow to obtain the cumulative long-term average monthly base-flow.
- Select the most stable (nearly linear) segment to obtain the slope of the stable base-flow.
- Using linear interpolation over the remaining months, the annual average base-flow is finally obtained.

The base-flow index ($B_p$) can be calculated by the following equation using the base-flow and the stable base-flow, with ($Q_b$) the base-flow and (Q) the total stream runoff, as in Equation (6).

$$B_p = \frac{Q_b}{Q} \tag{6}$$

Of the two gauging stations in the catchment area, data were collected for 2019 from the Nibéhibé station and for 2019–2020 from the Sikaboutou station.

3.2.2. WTF Method

The water table fluctuation method is based on the assumption that a rise in groundwater level in an unconfined aquifer is due to recharged water reaching the groundwater

table. This method requires data about specific yield and changes in water levels over time. The advantages of this approach include its simplicity and its insensitivity to the mechanisms by which water moves through the unsaturated zone [7,32]. However, it is difficult to determine a representative value for a specific yield and to ensure that the fluctuation in groundwater levels is a result of recharge, and not a result of changes in atmospheric pressure, the presence of entrapped air, or anthropogenic effects. Direct recharge is calculated using Equation (7):

$$R = S_y \times \frac{\Delta H}{\Delta t} \tag{7}$$

where R = recharge; $S_y$ = specific yield; $\Delta H$ = change in water table elevation (water level rise), and $\Delta t$ = the time period.

Specific Yield

To apply the WTF method, an estimation of the specific yield (Sy) is required [8]. Methods commonly used to determine Sy are laboratory methods, aquifer tests, water-budget methods, and the water table response to recharge [9]. In this study, the specific yield was estimated from pumping tests carried out in 134 boreholes distributed over the study area.

Determination of $\Delta H$: Water Level Rise

In this study, $\Delta H$ was defined as the difference between the maximum piezometric level and the assumed lowest piezometric level over a given period [7]. One option would be to calculate the water-level rise by subtracting the minimum water level from the maximum value during the recharge period. However, this does not reflect the natural recharge process. A better option agreed by most authors is to extrapolate the water table recession assuming that no recharge happens in this period. This is represented by the master recession curve (MRC) [10]. By calculating the distance from the peak (high water level) to this line (low water level), $\Delta H$ is obtained. This is probably the best technique to estimate groundwater recharge with the WTF method. The MRC can be derived by collecting all recession segments from all groundwater level graphs. Finally, a nonlinear regression model is fitted to the data. To exclude the effect of the Lobo River on the groundwater level rise, the distance at which the river level rise does not affect the groundwater level was calculated. Thus, the relationship between the groundwater table and the river water level in a steady state is defined by Equation (8) [33]:

$$y_x^2 = h_x^2 + \left(y_1^2 - h_1^2\right)\frac{L - x}{L} + \left(y_2^2 - h_2^2\right)\frac{x}{L} \tag{8}$$

where $h_1$, $h_x$, and $h_2$ are the initial water levels of the river (in m) at distance x (in m) and distance L from the river, and $y_1$, $y_x$, and $y_2$ are the water levels at the time of the flood (in m) at distance x and distance L from the river (Figure 8).

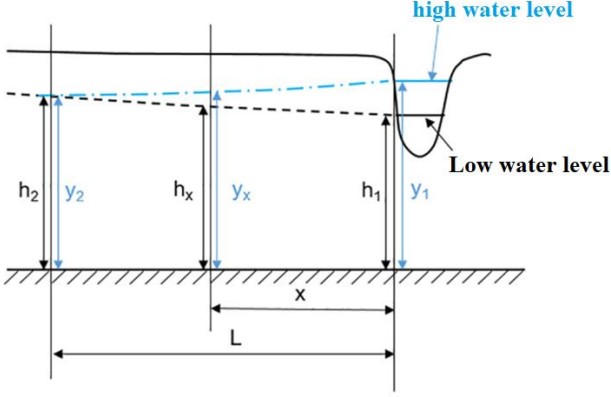

**Figure 8.** Impact of surface water rise on groundwater, modified from [33].

Considering distance L, at which the water level rise in the Lobo River does not affect the water table, with $y_2$ being equal to $h_2$, Equation (8) becomes Equation (9).

$$y_x^2 = h_x^2 + \left(y_1^2 - h_1^2\right)\frac{L - x}{L} \tag{9}$$

Considering the water levels in the Lobo River and the observation boreholes, distance *L* is calculated with Equation (10). All boreholes used for recharge estimation in the WTF method must be located at a distance greater than distance L from the river. Twenty-three boreholes meeting these conditions were finally selected (Table 1). This study is based on regular monitoring observations of piezometric data collected at these points between 2019 and 2020.

$$L = \frac{x\left(y_1^2 - h_1^2\right)}{\left(y_1^2 - y_x^2\right) - \left(h_1^2 - h_x^2\right)} \tag{10}$$

**Table 1.** Variations in piezometric level in 2019 and 2020 in the boreholes selected for the direct recharge study.

| Drillings | Water Level Fluctuation (m) | | |
|---|---|---|---|
| | 2018 | 2019 | 2020 |
| Tiahouo | 0.78 | 1.2 | 0.2 |
| Bazra-Nattis | 1.3 | 1.1 | 0.5 |
| Teneforo | 0.5 | 1.2 | 0.7 |
| Sokoura | 0.2 | 0.1 | 0.4 |
| Dananon | 0.3 | 0.6 | 0.5 |
| Vaafla | 0.8 | 1.4 | 0.3 |
| Seitifla | 1.2 | 2.8 | 0.8 |
| Diafla | 0.2 | 0.2 | 0.3 |
| Pelezi | 0.4 | 1.1 | 0.6 |
| Zoukouboue | 2.5 | 3 | 0.3 |
| Monoko-Zohi | 0.6 | 0.4 | 0.3 |
| Bohinou | 0.2 | 1.6 | 0.3 |
| Yacouba | 0.1 | 0.7 | −0.9 |
| Banoufla (Bediala) | 0.3 | 0.8 | 0.6 |
| Gnamienkro2 | 0.3 | 0.7 | 0.2 |
| Bonoufla (vavoua) | 1 | 0.9 | 0.6 |
| Ketro-Bassam | 2.3 | 1.9 | 2.1 |
| Broukro | 1.8 | 2.1 | 3.2 |
| Zouzoukro | 1.2 | 0.9 | 2.2 |
| Gbena | 1.4 | 0.9 | 1.9 |
| Dediafla2 | 2.4 | 1.4 | 3.5 |
| Vrouo1 | 0.4 | 1.3 | 0.8 |
| Bouhitafla | 0.8 | 1.2 | −1 |

### 3.2.3. Empirical Formulas

Several empirical formulas have been developed to estimate groundwater recharge. Some commonly used formulas are presented in Table 2. One of these empirical formulas for estimating recharge is based on rainfall (Equation (11)) [34]. This formula was later

adjusted by Baweja and Karanth [35] to give Equations 12 and 13. Other authors also developed formulas to estimate groundwater recharge (Equations (14)–(17)). To facilitate the calculation of recharge in our area, these empirical formulas were adjusted and were based on the one developed by [10], whose study area has similar characteristics to ours.

**Table 2.** Overview of the empirical equations for estimating groundwater recharge. Modified from [10]. R represents recharge (mm), P represents annual precipitation (mm), and MAP represents mean annual precipitation.

|  | **Modified Formula** | **Equation No.** |
|---|---|---|
| Chaturvedi [34] | $R = 3(P - 15)^{0.4}$ | (11) |
| Irrigation Research Institute, Roorkee [35] | $R = 2(P - 14)^{0.5}$ | (12) |
| Sehgal [36] | $1.8(P - 0.6)^{0.5}$ | (13) |
| Krishna Rao [37] | $0.37(P - 600)$ | (14) |
| Maxey-Eakin [38] | $0.22 \times P$ | (15) |
| Kirchner [39] | $0.26(MAP - 200)$ | (16) |
| Bredenkamp [40] | $0.29(MAP - 360)$ | (17) |

The mean absolute percentage error (MAP) was used to measure the accuracy of the calculations. MAP expressed as a percentage is defined by Equation (18) [41]:

$$\text{MAP (\%)} = \frac{100}{m} \sum_{i=1}^{m} \left| \frac{R_i - F_i}{R^i} \right| \tag{18}$$

where $R_i$ is the groundwater recharge estimated by WTF, and $F_i$ is the recharge calculated by the modified empirical formula. The lower the MAP, the more accurate the result.

3.2.4. Estimation of Annual Recharge

From the water balance, the annual groundwater recharge can be estimated using Equation (19) in the approach developed by [42]. According to this equation, all water reaching the water table either leaves the catchment as groundwater, flows to the surface, evaporates, or is stored:

$$R_a = \left( Q_{off}^{gw} - Q_{on}^{gw} \right) + Q^{bf} + ET^{gw} + \Delta S^{gw} \tag{19}$$

where $R_a$ is annual recharge, $Q_{off}^{gw}$ is groundwater flow out of the catchment, $Q_{on}^{gw}$ is groundwater flow into the catchment, $Q^{bf}$ is base-flow, $ET^{gw}$ is evapotranspiration from the groundwater table, and $\Delta S^{gw}$ is change in groundwater storage.

In this study, the Lobo catchment was considered a closed system, and the river was the main groundwater outlet. With the evapotranspiration of the water table considered to be zero in the area because of the depth of the water table, which can reach 30 m in some places, Equation (19) becomes Equation (20):

$$R_a = \Delta S^{gw} + Q^{bf} \tag{20}$$

where $R_a$ is annual recharge, $\Delta S^{gw}$ is change in groundwater storage or direct recharge estimated by the WTF method, and $Q^{bf}$ is base-flow estimated by WHAT.

**4. Results and Discussion**

*4.1. Recession Curve Displacement*

The recession curve displacement method and flow data from the gauging stations of Sikaboutou and Nibéhibé were used to estimate the groundwater recharge in our study area. Recharge calculated at the Sikaboutou station was 23.4 mm in 2019 and 11 mm in

2020. Between 2019 and 2020, a difference of 12.4 mm was observed at this station. This difference could be explained by the distribution of rainfall in the area. Indeed, as the Lobo catchment is quite large (about 7000 km$^2$), the distribution of rainfall may not be uniform over the area. This uneven rainfall distribution could make the recession curve displacement method unsuitable for our area.

At Nibéhibé station, the recharge was 8 mm in 2019. In 2020, it could not be estimated due to a lack of data. Studies in similar environments [8,10,29] have shown that the size of the study area could have an impact on the results obtained with this method. When the catchment area is quite large, as is the case in our study area (about 7000 km$^2$), the distribution of precipitation is not uniform, thereby making this method unsuitable. This difference between the recharge obtained at these two stations could also be explained by the presence of the Lobo reservoir downstream from the Sikaboutou station and upstream from the Nibéhibé station as well. Indeed, this reservoir retains a large quantity of water and reduces the downstream flows, which could explain the low values obtained at the Nibéhibé station. These values of recharge obtained by the displacement recession curve method are subject to the assumption inherent in this method. This assumption states that the recharge occurs instantaneously and uniformly, directly after a rainstorm [22], which is not the case, especially for large watershed areas. Therefore, this method is unsuitable for estimating recharge in large areas.

### 4.2. Base-Flow Analyses

Base-flows estimated by the constant slope method and the Automated Web GIS-Based Hydrograph Analysis Tool (WHAT) are presented in Table 3. Base-flows at Sikaboutou gauging station varied between 24.6 mm and 56.5 mm in 2019, with a mean of 42.6 mm. However, in 2020, they varied between 18.7 mm and 27 mm, with a mean of 23.4 mm. At the Nibéhibé gauging station, base-flow varied between 13.8 mm and 7.6 mm, with a mean of 10.8 mm. The base-flow calculated for the Sikaboutou station in 2020 is lower than that in 2019. The results of the Eckhardt separation method are closer to the graphical method. This may be due to the use of two parameters ($\alpha$ and BFImax) in the digital filter base-flow, where the BFImax parameter takes the physical properties of the basin into account. This is mainly due to the low rainfall recorded that year: in 2019, the average rainfall in the catchment area was 1237 mm, whereas in 2020, it was 742 mm, i.e., a drop of 495 mm.

**Table 3.** Recharge values estimated by the different hydrograph analysis methods.

| | Gauging Stations | | |
| --- | --- | --- | --- |
| | Sikaboutou | | Nibéhibé |
| | **2019** | **2020** | **2019** |
| Methods | | | |
| Graphical method (mm) | 24.6 | 18.7 | 7.6 |
| Local minimum method (mm) | 62.4 | 25.8 | 13.2 |
| Single parameter filter (mm) | 56.5 | 21.9 | 13.8 |
| Eckhardt method (mm) | 26.9 | 27 | 8.7 |
| Mean base-flow (mm) | 42.6 | 23.4 | 10.8 |

Table 4 shows a Spearman correlation matrix between the results of the different hydrograph analysis methods for each gauging station. The results of the different analysis techniques show a strong correlation (value close to 1) for both gauging stations. This means that all the methods can be considered appropriate for our area and confirms that they can be applied to estimate groundwater recharge.

**Table 4.** Spearman correlation coefficients between base-flow analysis results for both gauging stations.

| Methods | Graphical Method | Local Minimum Method | Single-Parameter Filter | Eckhardt Method |
|---|---|---|---|---|
| Sikaboutou station | | | | |
| Graphical method | 1 | | | |
| Local minimum method | 0.97 | 1 | | |
| Single-parameter filter | 0.98 | 0.99 | 1 | |
| Eckhardt method | 0.96 | 0.97 | 0.98 | 1 |
| Nibéhibé station | | | | |
| Graphical method | 1 | | | |
| Local minimum method | 0.99 | 1 | | |
| Single-parameter filter | 0.94 | 0.99 | 1 | |
| Eckhardt method | 0.98 | 0.99 | 0.99 | 1 |

*4.3. Direct Recharge Estimated by the WTF Method*

The specific yield, $S_y$, obtained by conducting pumping tests in 134 drillings, was 4.8%.

The groundwater direct recharge estimated by the water table fluctuation method depends on the maximum (November) and minimum (February) groundwater levels for the period (2018–2020). Figure 9 shows the variation in groundwater levels in the observation wells in the Lobo catchment. Analysis of the various piezometric records shows a rapid variation in the water table during the rainy season and a rapid drop during the dry season in all the boreholes in the study area. Overall, the water table changed very little during the study period, except for a few boreholes that experienced increases of up to 3 m. In the Lobo catchment, the water table can be very sensitive to direct recharge by rainfall due to its shallow depth.

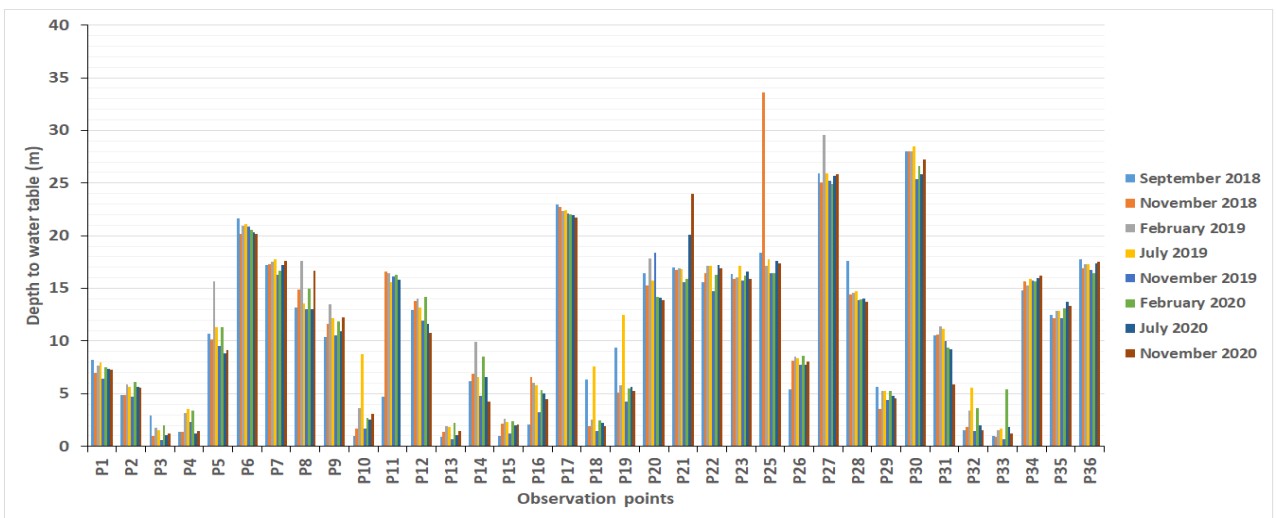

**Figure 9.** Depth to the water table in observation wells before, during, and after the wet season.

Groundwater direct recharge was estimated at each abandoned borehole acting as a piezometer. The WTF method was used to determine the actual recharge of the water table in the years 2018 and 2020, and the values obtained are shown in Table 1. Analysis of Table 1 shows that piezometric fluctuations were either positive or negative. Positive fluctuations representing an increase in water level in the aquifer varied between 0.1 m at Sokoura and 3 m at Zoukouboue and generated local recharges between 6.2 and 143.4 mm/year. In the Lobo catchment, some boreholes record direct recharges of over 100 mm/year.

The recharges calculated for these boreholes are significantly higher than those for other boreholes. This might be because heavy rain, or even light rain events, can affect the groundwater in the vicinity of these boreholes. Thus, the cumulative direct recharge is higher in this zone. The highest water table rises were observed in the abandoned boreholes of Zoukouboue, Seitifla, Broukro, Ketro-Bassam, and Bohinou, with values of 3, 2.8, 2.1, 1.9, and 1.6 m, respectively. In the Lobo catchment, all 23 boreholes used to calculate direct recharge experienced increased in piezometric levels during 2018 and 2019. In the region, the average direct recharge values for 2018 and 2019 were 44 mm and 57.3 mm, respectively, or 4% and 5% of the estimated 1081 and 1237 mm of rainfall. This recharge estimated by the WTF method is different from that estimated by the hydrograph method. This difference is due to the fact that each method uses different algorithms and depends on the assumptions made. The estimated average direct recharge in 2020 was 43 mm, generated by water levels in the boreholes that varied from 0.2 m to 3.5 m, respectively.

The largest fluctuations were recorded in Ketro-Bassam, Zouzoukro, Broukro, and Dediafla2, with values of 2.1; 2.2; 3.2, and 3.5 m, respectively. Negative fluctuations corresponding to a drop in the water level in the aquifer were found for the boreholes in Yacouba and Bouhitafla. The low number of boreholes with negative fluctuations (which represent a drop in the water table) in the catchment area is evidence of groundwater renewal. The mean direct recharge for the year 2020 was 43 mm, or 6% of the estimated 741.6 mm of precipitation. In the study area, the recharge decreased by 14.4 mm between 2019 and 2020, but between 2018 and 2019, it increased by 8.5 mm. The maximum and minimum values of recharge from 2018, 2019, and 2020 and their 75% and 25% quartiles are presented in Table 5.

**Table 5.** Estimated direct groundwater recharge in 2018, 2019, and 2020.

| Drillings | Direct Recharge (mm) | | |
|:---:|:---:|:---:|:---:|
| | **2018** | **2019** | **2020** |
| Tiahouo | 37.9 | 58.3 | 12.4 |
| Bazra-Nattis | 65 | 55 | 25.3 |
| Teneforo | 23.9 | 57.4 | 36 |
| Sokoura | 12.4 | 6.2 | 21 |
| Dananon | 14.1 | 28.2 | 23.4 |
| Vaafla | 38.3 | 67 | 15.3 |
| Seitifla | 58 | 135 | 40 |
| Diafla | 10 | 10 | 14 |
| Pelezi | 19.3 | 53 | 27.2 |
| Zoukouboue | 119.5 | 143.4 | 15.3 |
| Monoko-Zohi | 25.5 | 17 | 15 |
| Bohinou | 9.5 | 76 | 13 |
| Yacouba | 4.8 | 33.5 | 0 |
| Banoufla (Bediala) | 14 | 37.3 | 28.2 |
| Gnamienkro2 | 14.7 | 34.4 | 11 |
| Bonoufla (vavoua) | 45.6 | 41 | 31 |
| Ketro-Bassam | 111.7 | 92.3 | 100.4 |
| Broukro | 87.7 | 102.3 | 155 |
| Zouzoukro | 55 | 41 | 106.1 |
| Gbena | 65.5 | 42.1 | 93 |

**Table 5.** *Cont.*

| Drillings | Direct Recharge (mm) | | |
|---|---|---|---|
| | **2018** | **2019** | **2020** |
| Dediafla2 | 117.3 | 68.4 | 167 |
| Vrouo1 | 18.5 | 60 | 36 |
| Bouhitafla | 38.9 | 58.4 | 0 |
| Mean groundwater recharge | 44 | 57.3 | 43 |
| Maximum | 119.5 | 143.4 | 167 |
| Minimum | 4.8 | 6.2 | 0 |
| 75% Quartile | 65 | 68 | 38 |
| 25% Quartile | 14.1 | 36 | 14.3 |

These results are contrary to those of [43], who used the hydrological balance method to estimate a direct recharge of 165 mm/year at Nibéhibé in the Lobo catchment over the period 2007–2020, i.e., about 12% of the rainfall. This difference could be explained by the fact that the direct recharge estimate based on groundwater data could present better results than methods using rainfall data, which can overestimate this parameter [43]. This direct recharge value is a function of rainfall and the number of points used to estimate the parameter but is also a function of specific yield. Indeed, the direct recharge estimated with the assumption of a uniform specific yield over the whole area could be either under- or overestimated. The size of the area with unevenly distributed rainfall could also have an impact on this parameter [8].

*4.4. Empirical Formulas*

Results of the groundwater recharge calculated using the empirical formulas are presented in Table 6. The recharge estimated with empirical formulas modified from [10] was between 0 and 272.1 mm in 2019 and between 0 and 163.1 mm in 2020. This large difference between recharge values could be explained by the empirical formulas, which are valid for a particular area only. The formulas modified from [10] generally result in higher recharge values compared with those calculated with the WTF method. A comparison of the results obtained using the two methods shows that the equations of Sehgal, Maxey–Eakin, and Krishna Rao (Irrigation research Institute, Roorkee) overestimate the recharge, whereas the equations of Chaturvedi, Bredenkamp, and Kirchner underestimate the recharge in the Lobo River catchment. These observations are also in agreement with those of [10] and can be explained by the equations based on precipitation only. To obtain empirical formulas adapted to our area and that can be used for future studies, the described equations required modification (Table 7). Once the formulas had been modified and adapted to our area, they provided a better fit to our study area, with recharge values ranging from 59.7 to 67.9 mm in 2019 and from 35.7 to 50.2 mm in 2020. The MAP are between 4.5% and 18.8% for 2019 and between 8.2% and 17.5% for 2020. These values are close to the direct recharge estimated by the WTF method in 2019 and 2020. These rather low MAP values indicate that the modified formulas can be applied to our study area, and that the different equations could be used in the future to estimate recharge in our study area. The recharge values estimated with empirical formulas are not sufficiently constant, because the formulas use only one parameter.

**Table 6.** Empirical formulas modified according to [10].

| Original Authors | Recharge 2019 | Recharge 2020 | MAP (%) 2019 | MAP (%) 2020 |
|---|---|---|---|---|
| Chaturvedi | 51.5 | 41.8 | 9.9 | 26.8 |
| Irrigation Research Institute, Roorkee | 69.9 | 53.9 | 22.3 | 5.64 |
| Sehgal | 63.2 | 49 | 10.70 | 14.3 |
| Krishna Rao | 235.7 | 52.4 | 312.7 | 8.3 |
| Maxey-Eakin | 272.1 | 163.1 | 376 | 185.4 |
| Kirchner | 0 | 0 | 100 | 100 |
| Bredenkamp | 0 | 0 | 100 | 100 |

**Table 7.** Modified empirical formulas for the Lobo catchment.

| Original Authors | Modified Formulas | Recharge 2019 | Recharge 2020 | MAP (%) 2019 | MAP (%) 2020 |
|---|---|---|---|---|---|
| Chaturvedi | $1.21\,(P-15)^{0.4}$ | 61.6 | 50.2 | 7.8 | 17.5 |
| Irrigation Research Institute, Roorkee | $1.8\,(P-14)^{0.5}$ | 62.9 | 48.5 | 10.1 | 13.5 |
| Sehgal | $1.7\,(P-0.6)^{0.5}$ | 59.7 | 46.2 | 4.5 | 8.2 |
| Krishna Rao | $0.055\,(P-55)$ | 65 | 37.7 | 13.7 | 11.6 |
| Maxey-Eakin | $0.051 \times P$ | 63.1 | 37.8 | 10.3 | 11.5 |
| Kirchner | $0.69\,(MAP-10)$ | 64.2 | 35.7 | 12.3 | 16.3 |
| Bredenkamp | $0.73\,(MAP-10)$ | 67.9 | 37.8 | 18.8 | 11.5 |

The results obtained by modifying those formulas are acceptable if we take into account the clayey nature of the surface layer of the catchment with its low porosity. Groundwater recharge accounts for about 5% of precipitation in the area in 2019 and 6% in 2020, whereas the major proportions of precipitation correspond to evapotranspiration and runoff [43]. Sustainable exploitation of this resource should not exceed the estimated recharge in this catchment.

*4.5. Annual Recharge in the Lobo Catchment*

The mean annual recharge in the Lobo catchment, including the diffuse or direct recharge estimated by the WTF method and the mean base-flows over the years 2019 and 2020, varied from 84 to 66.4 mm, with a difference of 17.6 mm in the mean annual recharge in one year. This annual recharge is dominated by direct rainfall recharge, which accounted for about 68% of the annual recharge in 2019 and about 65% in 2020. The direct recharge estimated by the WTF method was generally higher than that estimated with the base-flow methods. This difference is due to the fact that each method uses different algorithms and depends on the assumptions made [9]. This variation in the direct groundwater recharge estimate between the WTF and hydrograph analysis methods is because the recharge estimated by hydrograph analysis takes into account diffuse recharge and recharge through riverbed leakage, whereas the WTF method only takes into account diffuse recharge and is less affected by indirect recharge [9]. The recharge obtained by the WTF method may not represent the recharge of the whole catchment area, as the monitoring wells do not cover the whole area.

**5. Conclusions**

Groundwater recharge in the study area was estimated using several methods. With the WTF method, it was estimated at 44 mm and 57.3 mm in 2018 and 2019, respectively, representing 4% and 5% of precipitation, and at 43 mm in 2020, representing about 6% of precipitation. Although all methods adapted to our area provided similar results, each

method had its capacities, limitations, and uncertainties. If the quality of the data was better (e.g., for the hydrographic analyses), the results would be more accurate. The recharge calculated recession curve displacement method at the Sikaboutou station was 23.4 mm in 2019 and 11 mm in 2020. At Nibéhibé station, the recharge was 8 mm in 2019. The empirical formulas are not valid for all areas: the constant slope method is not suitable for the rainy season (overestimates base-flows), and WHAT depends on the river flows and filters used. The average annual base-flow at Sikaboutou station was 42.6 mm in 2019 and 23.4 mm in 2020. At Nibéhibé station, the base-flow was 10.8 mm in 2019. The average annual recharge of the area, including the direct recharge estimated by WTF and the base-flows estimated by WHAT, varied between 84 mm in 2019 and 66.4 in 2020. In this study, the most reasonable estimates of direct recharge were obtained with the WTF method. The modified empirical formulas for the Lobo catchment could be used for future estimation of recharge, as long as radical changes in land use and climate are not observed. However, a more integrated approach to these methods with a larger data set could yield interesting results.

**Author Contributions:** The completion of this work was possible thanks to its contributors. K.K.J.O. wrote the protocol, collected the data, processed the data, and wrote the manuscript. D.B., M.o.M.J., and E.S.G. assisted in the analysis of the data and research in the interpretation for the results. P.F. and G.D. managed the literature research and supervised the work. All authors have read and agreed to the published version of the manuscript.

**Funding:** This research received no external funding.

**Institutional Review Board Statement:** No applicable.

**Informed Consent Statement:** No applicable.

**Data Availability Statement:** No applicable.

**Acknowledgments:** The data used in this study were collected by our research team for hydrogeological data. The hydrological data used in this study were collected by the hydrology team of the Environmental Science and Technology Laboratory of the Jean Lorougnon Guede University, Daloa, whom we would like to thank.

**Conflicts of Interest:** The authors declare no conflict of interest.

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
