# Peer review of "Estimation of Groundwater Recharge in the Lobo Catchment (Central-Western Region of Côte d’Ivoire)"

_hydrology, doi:10.3390/hydrology9020023_

Round 1
Reviewer 1 Report
This is an interesting study. However, it needs several improvements:
- The manuscript needs linguistic improvement and scientific style, such as the first sentence in introduction section is not impressive…
- Most of the figures are not explained well and the figure resolution not good i.e., In Fig.1, font size of legend is too large and the figure is blurred and not explained well; font size problem in Fig.2; Figure 4 is important but it has no explanation, please explain this fig and reorganize your paper accordingly; similarly, please revise other figs
- You have used different methods to estimate groundwater recharge, that’s great; however, it is only for two years. I strongly suggest the authors to extend your paper to the recharge of 3-5 years.
- The results need better description
- Please add main conclusions from your study
Reviewer 2 Report
Recharge is a very important process to adequately capture in order to sustainably utilize groundwater, but is notoriously difficult to ascertain. The study design of this paper attempts to utilize several methods in order to capture the variability of groundwater recharge rates for the Lobo catchment. The background on the various methods is adequate, with enough presentation of the methods and the underlying equations for each method. However, there is not enough presentation of the actual data that is used for this paper. If my understanding is correct, the piezometric data is only seasonal, which can make it difficult to come up with recharge estimates at such a coarse resolution. Based on Healy and Cook (2002; Hydrogeology J., vol. 10, p. 91-109), the most appropriate time frame for the application of WTF methods are hours or a few days as WTF methods are best when identifying individual events in order to identify the recharge events over the course of the year. This was applied for the river hydrograph separation, so it is difficult to make comparisons between daily hydrograph and seasonal piezometer data.
Some further editorial comments for improvements:
(39-43): Sustainable management of groundwater should start with usage not exceeding recharge levels. However, any usage will come at the cost of lower discharge levels, so that will have a downstream effect on indirect recharge.
(45-50): Sentence is repeated twice.
(84-86): Figure 2, please make the legend consistent by labeling the years as 2019, rather than 19. Also, make the caption clear that Q is discharge.
(98-99): Figure, 3, the patterns are a little difficult to distinguish, particularly the biotite heterogenous granite and the schist.
(106): Figure 4 looks good -- this is a good presentation of recharge processes.
(340-342): Figure 9, at this scale it is somewhat difficult to see the differences in water levels. Since this is one of the few presentations of the data, it makes it difficult to make judgments on whether the data is adequate to support WTF methods.
(366-368): Two years of recharge is not enough to begin make judgments about the root cause of the underlying cause of changes in recharge rates, particularly at such a large catchment scale.
Round 2
Reviewer 1 Report
I think the manuscript in present form is acceptable for publication.
Reviewer 2 Report
Thank you for making the corrections. I do not have any other comments at this time.